# Bioactivity of Microencapsulated Cell-Free Supernatant of *Streptococcus thermophilus* in Combination with Thyme Extract on Food-Related Bacteria

**DOI:** 10.3390/foods13020329

**Published:** 2024-01-20

**Authors:** Esmeray Kuley, Nagihan Kazgan, Yetkin Sakarya, Esra Balıkcı, Yesim Ozogul, Hatice Yazgan, Gülsün Özyurt

**Affiliations:** 1Department of Seafood Processing Technology, Faculty of Fisheries, Cukurova University, 01330 Adana, Turkey; ekuley@cu.edu.tr (E.K.); nghn.kzgn@gmail.com (N.K.); ysakarya@cu.edu.tr (Y.S.);; 2Department of Gastronomy and Culinary Arts, Faculty of Tourism, Yozgat Bozok University, 66900 Yozgat, Turkey; esra.balikci@bozok.edu.tr; 3Department of Food Hygiene and Technology, Faculty of Ceyhan Veterinary Medicine, Cukurova University, 01960 Adana, Turkey; hyazgan@cu.edu.tr

**Keywords:** *Streptococcus thermophilus*, thyme extract, antimicrobials, foodborne bacteria

## Abstract

The bioactive properties of the combination of microencapsulated cell-free supernatant (CFS) from *Streptococcus thermophilus* and thyme extract on food-related bacteria (*Photobacterium damselae, Proteus mirabilis*, *Vibrio vulnificus, Staphylococcus aureus* ATCC29213, *Enterococcus faecalis* ATCC29212, and *Salmonella* Paratyphi A NCTC13) were investigated. The microencapsulated CFS of *S. thermophilus,* in combination with ethanolic thyme extract, had a particle size in the range of 1.11 to 11.39 µm. The microencapsulated CFS of *S. thermophilus* had a wrinkled, spherical form. In the supernatant, especially at 2% (*v/w*), the thyme extract additive caused a decrease in the wrinkled form and a completely spherical structure. A total of 11 compounds were determined in the cell-free supernatant of *S. thermophilus*, and acetic acid (39.64%) and methyl-d3 1-dideuterio-2-propenyl ether (10.87%) were the main components. Thyme extract contained seven components, the main component being carvacrol at 67.96% and 1,2,3-propanetriol at 25.77%. Significant differences (*p* < 0.05) were observed in the inhibition zones of the extracts on bacteria. The inhibitory effect of thyme extract on bacteria varied between 25.00 (*P. damselae*) and 41.67 mm (*V. vulnificus*). Less antibacterial activity was shown by the microencapsulated CFS from *S. thermophilus* compared to their pure form. (*p* < 0.05). As a result, it was found that microencapsulated forms of CFS from *S. thermophilus,* especially those prepared in combination with 2% (*v*/*w*) thyme extract, generally showed higher bioactive effects on bacteria.

## 1. Introduction

Fish meat has a high nutritional content and contains essential macro- and micronutrients such as protein, fats, vitamins, and minerals. In addition, since fish muscles have a high post-mortem pH (>6), a high concentration of low-molecular-weight compounds, and high water activity, microbial activity is the primary cause of raw fish deterioration [1,2]. Raw seafood products can easily spoil and also get infected by pathogens, including bacteria, naturally or via cross-contamination. *Staphylococcus aureus*, *Salmonella*, *E. coli*, *Vibrio parahaemolyticus*, *Aeromonas hydrophila*, *Clostridium botulinum*, *Listeria monocytogenes*, *Yersinia* spp., and *Enterococcus faecalis* are common pathogens linked to human foodborne diseases through the consumption of contaminated seafood products [3,4,5,6]. Various synthetic antimicrobials are used to control pathogen growth in food products, but these antimicrobials cause consumer concern due to their harmful effects. For this reason, studies on the use of natural alternative antimicrobials of plant, animal, or bacterial origin are gaining importance.

*Streptococcus thermophilus* is a member of the lactic acid bacteria that are generally recognised as safe (GRAS) in the United States and as having a “qualified presumption of safety” (QPS) in the European Union because of their long history of safe usage in foodstuff [7]. *S. thermophilus* is naturally found in fermented milk, cheese, other dairy products, and some plants [8]. Its main feature in fermentation is the formation of lactic acid, which results in rapid acidification and the suppression of other microorganisms. The secondary metabolites generated by *S. thermophilus*, acetaldehyde and exopolysaccharides, have a significant impact on the product’s flavour and texture. [9]. Along with lactose, *S. thermophilus* also has the ability to ferment sucrose, glucose, galactose, and sugar. Due to their capacity to rapidly ferment lactose, *S. thermophilus* is frequently utilised symbiotically with other LAB members as starter cultures in fermented milk products. *S. thermophilus* has been known to possess probiotic potential due to its immunomodulatory, antibacterial, antioxidant, and anti-inflammatory properties [10,11]. This member is among the most important industrial starter cultures after *Lactococcus lactis* in dairy products [12]. Some members of *S. thermophilus* can produce thermophilins, which are short peptides that have inhibitory effects on Gram-positive pathogenic *Enterococcus faecalis*, *Clostridium botulinum*, *Staphylococcus aureus*, and *Listeria monocytogenes* [8].

A member of the *Lamiaceae* family of plants, thyme is used in lotions, perfumes, and other cosmetic items, as well as food and beverage and confectionery products as a flavour enhancer. Thyme is well-liked as a medicinal herb and food preservative because of its antiseptic, bronchiolytic, antispasmodic, and antibacterial qualities [13]. These plants’ essential oils contain high concentrations of thymol, carvacrol, p-cymene, and -terpinene [14]. According to studies conducted in vitro, essential oils and the chemical components they contain, such as thymol and carvacrol, have antibacterial and antioxidant effects [15,16]. Besides these, the extracts from thyme may be used as antibacterial and anticarcinogenic agents to combat bacterial pathogens that cause food poisoning and nosocomial infections. [17]. The thyme extract demonstrated inhibitory activity on microbial growth, as well as bactericidal efficacy against *Listeria monocytogenes*, *E. faecalis*, *Bacillus cereus*, *E. coli*, *Salmonella enterica*, *S. aureus*, and *Yersinia enterocolitica*, at relatively low concentrations [18].

The food industry uses the encapsulation method to extend the shelf life of various food products by including bioactive ingredients and shielding them from unfavourable environmental factors [19]. In addition, the application of microencapsulation in the food business offers a number of benefits. Among them are lowering the main ingredient’s reactivity to environmental factors, slowing the rate at which an external component is absorbed into the main ingredient’s material, facilitating easier handling through encapsulation, regulating the main ingredient’s material’s release, covering up the taste of the main ingredient, and diluting the main ingredient material when used in very small amounts [20,21]. The purpose of the encapsulation approach is to guarantee that the active components are kept in the coating material at nanometric, micrometric, or millimetric dimensions [22]. The most popular encapsulation techniques are spray drying, freeze drying, extrusion, coating with an air suspension, applying liposomes, and crystallisation, with capsule sizes ranging from a few micrometres to a few centimetres [23,24].

It has been noted that spray drying significantly reduces the antioxidant activity after microencapsulation because the major phenolic components of thyme extract are impacted by high temperatures in the microencapsulation process [25]. Furthermore, Ozyurt et al. [26] observed that the amount of essential amino acids and gamma-linoleic fatty acids in the product was retained when maltodextrin was used in increasing amounts as a coating material during spray drying. Therefore, for each component that is intended to be coated, it is crucial to establish the proper microencapsulation settings, including device intake and outlet temperature, flow feed rate, wall material selection, and ratio.

In comparison to probiotic cells, supernatants have better antipathogenic effects and are more stable during storage, according to Saadatzadeh et al. [27]. This is why lactic acid bacteria supernatants are encapsulated, which opens up new antibacterial possibilities. The lactic acid bacteria’s cell-free supernatant (CFS) can be prepared in liquid form and is rich in bioactive compounds. The conversion of the obtained product into microencapsulated form will give the product a longer shelf life and make it simpler to apply in foods by preserving the product from external conditions, in addition to the inclusion of thyme extract, in order to boost the bioactivity of these supernatants. This study aimed to determine the antimicrobial properties of CFS from *S. thermophilus* and thyme (*Origanum vulgare*) extract combinations before and after microencapsulation.

## 2. Materials and Methods

### 2.1. Thyme Extract and Bacterial Strains

Ethanolic thyme (*Origanum vulgare*) extract (20 g) was obtained from Biomesi Bioagro Technology R&D (Adana, Turkey). *Streptococcus thermophilus* [28] was obtained from Kahramanmaraş Sütçü İmam University. *Photobacterium damselae*, *Proteus mirabilis*, and *Vibrio vulnificus* were isolated from spoiled anchovy, mackerel, and sardine meat [29]. Gram-positive *Staphylococcus aureus* (ATCC29213), *Enterococcus faecalis* (ATCC29212), and Gram-negative *Salmonella* Paratyphi A (NCTC13) were purchased from the National Collection of Type Cultures in London, United Kingdom, and the American Type Culture Collection in Rockville, Maryland, in the United States.

### 2.2. Methods

#### 2.2.1. Preparation of CFS from *S. thermophilus*

A modification of Lin and Yen’s method [30] was used to create lactic acid bacteria supernatant. M17 broth (56156, Sigma-Aldrich Chemie GmbH, Steinheim, Germany) was used to culture *S. thermophilus* for 48 h at 37 °C. After the supernatant was transferred to 15 mL falcon tubes, the supernatant was centrifuged at 8000 RPM for 10 min at 4 °C. The obtained supernatant was put through a membrane filter, exposed to UV light for 20 min, and kept at 4 °C until analysis.

#### 2.2.2. Microencapsulation Process

A modification of Marcela et al.’s [31] method was followed for microencapsulation. CFSs obtained from *S. thermophilus* were microencapsulated individually. Other groups were microencapsulated together with the CFS of *S. thermophilus* containing 1 and 2% thyme extract (*v*/*w*). All groups were coated with 25% maltodextrin (DE: 18–20) Alfasol, Turkey) before drying. The mixture was mixed with Ultra-Turrax for 10 min. A 250 mL measure of the sample from each group was dried using a mini spray dryer (Buchi-290, Flawil, Switzerland). Spray drying conditions were an inlet temperature of 130 °C, an outlet temperature of 75 °C, an aspiration rate of 30 m3/hour, and a feed rate of 20 mL/min. After spray drying, all samples were placed in light-proof plastic bottles and stored at 4 °C until the day of analysis.

#### 2.2.3. Microencapsulation Morphology

The morphology of the particles obtained through spray drying was characterized using Quanta 650 field emission scanning electron microscopy (SEM, FEI Company, Hillsborn, OR, USA) at Cukurova University Central Research Laboratory (CUMERLAB, Adana, Turkey). SEM images were obtained at room temperature and 20 kV voltage, and the results were visually recorded at 10,000 and 20,000 magnification after gold coating with a Q150R ES Coater (Quorum Technologies, Lewes, UK).

#### 2.2.4. GC–MS Analysis of Samples

The chemical composition of the thyme extract and CFS of *S. thermophilus* was determined using an Agilent 7890A gas chromatograph (GC) incorporating a mass spectrometer (MS, Agilent, Palo Alto, CA, USA). A non-polar (5% -phenyl)-methylpolysiloxane (30 m × 250 μm × 0.25 μm, Agilent 1901S-433HP-5MS) was used as column. The flow rate of helium benefited as the carrier gas was 1.5 mL/min. GC–MS conditions were set to 50 °C initial and 240 °C final oven temperature, 1 µL injection volume, and 105 min run time. The detection of peaks found in GC–MS was obtained through comparison with those in the commercial library of NIST, EPA, and NIH version 2.0.

#### 2.2.5. Antimicrobial Activity Assay of Extracts

##### Agar Well Diffusion Method

Mueller–Hinton Agar (MHA, Merck 1.05437, Darmstadt, Germany) was used to measure the in vitro antimicrobial activity level of non-encapsulated (pure) and encapsulated CFS from a combination of *S. thermophilus* and thyme extract (1%, *v*/*w*), in accordance with the well diffusion method of Hwanhlem et al. [32]. For the antimicrobial activity test, microencapsulated samples were dissolved in distilled water at a ratio of 1:2. Food-related bacteria (*Photobacterium damselae*, *Proteus mirabilis*, *Vibrio vulnificus*, *Staphylococcus aureus*, *Enterococcus faecalis,* and *Salmonella* Paratyphi A) were grown in Nutrient broth (Merck 1.05443, Darmstadt, Germany) at 37 °C for 24 h and standardised to 0.5 McFarland cell density (10^8^ cfu/mL). Each bacterial cell culture (100 µL) was inoculated into a Petri dish containing 20 mL of Mueller–Hilton agar. Five 5 mm wells were formed in a solid medium. A 50 µL measure of pure or microencapsulated extract was inoculated into 4 wells. Distilled water, or maltodextrin, was transferred to the other well as a control. Petri dishes were then incubated at 37 °C for 24 h. After incubation, the inhibition zones formed around each well were measured in mm using callipers.

##### Minimum Inhibitory (MIC) and Bactericidal Concentration (MBC)

Minimum inhibitory concentration (MIC) and minimum bactericidal concentration (MBC) were determined according to the microdilution method of CLSI [33]. Test microorganisms incubated at 37 °C for 24 h were standardised to 0.5 MacFarland cell density. Mueller–Hinton Broth (MHB, Oxoid, CM0405, Basingstoke, UK) was used as the medium. A 50 mg/mL stock solution prepared from pure and microencapsulated extracts was diluted to 0.19 μg/mL in sterile tubes. The tube containing only stock solution or pure culture was considered as the control, and the other tubes containing MHB contained test microorganisms and diluted stock solutions. The test tubes were prepared repeatedly and incubated at 37 °C for 24 h. Bacterial growth in the test tubes was compared with the control tubes, and the tubes with the lowest inhibition of bacterial growth were recorded as MIC. In line with the MIC results, the tubes without bacterial growth were inoculated onto the Mueller–Hinton Agar surface, the Petri dishes were incubated at 37 °C for 24 h, and MBK values were recorded.

#### 2.2.6. Statistical Analysis

One-way analysis of variance (ANOVA) and Duncan’s Multiple Comparison Test (SPSS 22, Chicago, IL, USA) were used for the statistical analysis. A significant difference between the groups was shown by the value of *p* < 0.05.

## 3. Results and Discussions

### 3.1. Morphology of Encapsules

Microencapsulations are defined as microparticles between 0.2 and 5000 μm in size [34]. In this study, the microencapsulated CFS from *S. thermophilus* had a particle size of 1.15–14.08 µm, the microencapsulated CFS from *S. thermophilus* combined with 1% (*v*/*w*) ethanolic thyme extract had a particle size of 1.11–11.50 µm, and the microencapsulated CFS of *S. thermophilus* combined with 2% (*v*/*w*) ethanolic thyme extract had a particle size of 2.71–11.39 µm (Figure 1). The particle sizes obtained through spray drying varied based on the type of coating, the quantity and consistency of the substance included, and the drying environment [35].

Microencapsulated CFS of *S. thermophilus* had a wrinkled, spherical form. Thyme supplementation in the supernatant caused a decrease in the wrinkled form. In particular, the microencapsulated CFS from *S. thermophilus* with a 2% (*v*/*w*) thyme additive was completely spherical (Figure 2). Differences in spray drying conditions can affect the spherical form of particles [36]. Unlike in this study, microencapsulated CFS from *Lactiplantibacillus plantarum* exhibited a more wrinkled form in the presence of propolis extract in [37]. The collapses occurring in microencapsules may be due to the possibility of moisture transport during the drying period [25,38].

### 3.2. Chemical Composition of CFS from S. thermophilus

The chemical composition of the CFS from *S. thermophilus* is displayed in Table 1. CFS from *S. thermophilus* included 11 different substances, but the predominant constituents were acetic acid (39.64%), methyl-d3 1-dideuterio-2-propenyl ether (10.87%), and 7-octen-2-ol, 2-methyl-6-methylene (10.46%) (Figure 2). Propane, 2-fluoro-2-methyl- (8.92%), imidodicarbonic diamide (7.64%), tetrahydro-1,3-oxazine-2-thione (7.21%), 1,2,3-propanetriol (5.8%), and 2-methyl-1,3-oxathiolany-propionic acid ethyl ester (5.42%) were other important components found in the CFS from *S. thermophilus*.

Organic acids (such as lactic acid, acetic acid and propionic acid), fatty acids, hydrogen peroxide, diacetyl, and bacteriocins are among the bioactive compounds produced by LAB that exhibit antimicrobial activity against pathogens and spoilage organisms [39]. The antimicrobial activity of the CFS of lactic acid bacteria is mainly due to lactic acid and acetic acid. Acetic acid, a product of carbon metabolism, shows broad-spectrum antimicrobial activity against bacteria, moulds, and yeasts due to its higher pKa value (4.87) compared to other acids [40,41]. Some LAB strains were able to degrade lactic acid into acetic acid. It has been reported that each mole of lactic acid was converted into approximately 0.5 mol of acetic acid, 0.5 mol of 1,2-propanediol, and traces of ethanol by certain LAB strains [42,43]. This may be the reason for the high proportion of acetic acid (39.64%) in the chemical composition of CFS from *S. thermophilus*. *Lactiplantibacillus plantarum* and *Lactobacillus acidophilus* produced 0.031–6.491 mg/mL lactic acid, 0.372–0.863 mg/mL protein, 0.009–0.029 mg/mL hydrogen peroxide, and 0.450–0.662 mg/mL diacetyl [44]. Bioactive substances produced by lactic acid bacteria may vary according to the cultural conditions and physiological characteristics of bacterial members [44].

Table 2 gives the chemical content of thyme extract. Thyme extract contained seven components, and the main components were phenol, 2-methyl-5-(1-methylethyl) (carvacrol) with 67.96%, and 1,2,3-propanetriol (glycerin) with 25.77%. Additionally, the extract of thyme had 3.97% of propane, 2-fluoro-2-methyl-methyl, and contained less than 0.5% of other components. According to Saleem et al. [45], carvacrol and thymol are the two primary constituents of fresh and dried plants, respectively. Carvacrol (52.32%), linalool L (15.96%), thymol (9.59%), 1-methylpyrroline (3.53%), and trans-caryophyllene (3.43%) were the major constituents of thyme (*Zataria multiflora*) extract [46]. Mehrabi et al. [47] reported that the main components in thyme were thymol (25.30%), δ-2-carene (8.825%), and carvacrol (8.43%). Mohammadigholami [48] reported that 45 components were found in thyme essential oil, and the main ones were carvacrol (52.4%), γ-trpynene (12.1%), and thymol (10.4%). Thymol (37.54%), p-cymene (14.49%), terpinene (11.15%), linalool (4.71%), and carvacrol (4.62%) were listed as the primary compounds of thyme essential oil by Fadil et al. [49]. The dominant compounds in thyme plants such as *Thymus migricus*, *Thymus eriocalyx*, *Thymus serpylum*, *Zataria multiflora*, and *Thymus kotschyanus* were linalool (41.8%), geraniol (61.8%), para cymene (23.8%), carvacrol (57.7%), and uulegone (37.2%) [50]. Thymol (45.16 %), phytol isomer (7.17 %), carvacrol (5.2 %), 9-octadecenal (4.84 %), caryophyllene oxide (4.27%), and trans-β-caryophyllene (2.86 %) were reported as components of thyme extract obtained through supercritical-flow CO_2_ extraction [51]. The main components identified in the ethanolic extract obtained from the leaves and branches of thyme (*Zataria multiflora*) were carvacrol (52.32%), linalool (15.96%), thymol (9.59%), 1-methylpyrroline (3.53%), and trans-caryophyllene (3.43%) [42]. Phytochemical compounds of *Thymus linearis* leaf extract determined using GC-MS were ethyl (9z, 12z)-9, 12-octadecadienoate (22.58%), palmitic acid (11.95%), ethyl palmitate (9.89%), (5.03%), stigmast-5-En-3-Ol, (3. Beta.)- (4.54%), (Z, Z)-6, 9-Cis-3, 4-epoxy-nonadecadiene (3.60%), carvacrol (3.59%), cryptomeridiol (3.22%), heptadecanoic acid, ethyl ester (2.03%), and naphthalene, decahydro-(1.28%) [52]. Various factors, such as geographical conditions, cultural differences, plant development stage, drying methods, and extraction methods, affect the qualitative and quantitative properties of bioactive components in a plant [46,47,53]. These variables are the cause of the detection of various components in thyme extract.

### 3.3. Antimicrobial Activity Analysis of Samples

#### 3.3.1. Inhibition Diameter Zone of the Samples

The inhibitory zones of the CFS from *S. thermophilus* on food-related bacteria before and after microencapsulation with the addition of thyme extract at various ratios are shown in Table 3. Significant differences (*p* < 0.05) were observed in terms of the inhibition zones of extracts on bacteria. The highest inhibition effect was observed in the group where the extract was directly applied (*p* < 0.05). The inhibitory effect of thyme extract on bacterial strains varied between 25.00 mm (*P. damselae*) and 41.67 mm (*V. vulnificus*). The inhibition zone of thyme extract on *Salmonella* Paratyphi A growth was 27.33 mm. Although thyme essential oil showed strong antibacterial activity against foodborne pathogenic bacteria such as *S. aureus* ATCC 9144, its extracts did not exhibit any antimicrobial activity [16]. Sabzikar et al. [46] reported that an ethanolic extract of thyme was highly active against the growth inhibition of *S. aureus* and *Candida albicans* and showed good antibacterial and antifungal activity. The highest level inhibitory zone diameters were observed in all concentrations of the ethanolic and acetone thyme extracts against *Salmonella* Typhi and *E. coli*, respectively. Conversely, the lowest level inhibitory zone diameters were observed in these extracts against *Bacillus cereus* and *Escherichia coli* [54].

With an inhibitory zone of 7.33 mm, *Photobacterium damselae* was the bacterium with the highest resistance to *S. thermophilus* extract. The most sensitive species to *S. thermophilus* extract was *V. vulnificus* (17.33 mm), as in thyme extract. Mehrabi et al. [47] reported that free cell extracts obtained from *Lactobacillus acidophilus* KMP, *Lactiplantibacillus plantarum* KMP, and *Pediococcus pentosaceus* KMP inhibited the proliferation of pathogenic *E. coli* in a time-dependent manner. The inhibition zones of *L. plantarum* KMP towards *E. coli* after 8, 10, 12, 14, and 16 h of incubation were 26.6, 24.9, 22.5, 20.3, and 17.9 mm, respectively. According to another investigation, *Micrococcus luteus* was inhibited by *Streptococcus macedonicus* free cell supernatant in a 13.5 mm inhibitory zone [55].

Microencapsulated *S. thermophilus* extracts were more ineffective against bacterial growth than non-microencapsulated forms (*p* < 0.05). Before or after microencapsulation, cell-free extract from *S. thermophilus* displayed the strongest inhibitory impact against *Vibrio vulnificus* and the lowest inhibition effect against *Photobacterium damselae*. Kuley et al. [37] found that *Proteus mirabilis* was most sensitive to the antimicrobial action of crude and microencapsulated CFS obtained from *Lactiplantibacillus plantarum*, with an inhibition zone between 8.33 and 8.00 mm.

#### 3.3.2. MIC and MBC of the Samples

Table 4 shows the MIC and MBC of non-microencapsulated and microencapsulated samples. The CFS from *S. thermophilus* exhibited a low inhibition concentration (12.5 and 25 mg/mL, respectively) against *V. vulnificus* and *Proteus mirabilis*. The antimicrobial activity of the free cell extract is due to lactic acid, acetic acid, long-chain fatty acids and esters, and proteinaceous compounds [41]. The CFS from *L. plantarum* has been reported to have a MIC value of 50 mg/mL against fish spoilage bacteria [37]. In the presence of CFS obtained from *Streptococcus salivarius* M18 bacteria, the proliferation of *P. aeruginosa* ATCC 27853 was reported to be reduced by approximately 60% after 6 h of incubation at 37 °C and almost entirely inhibited after 24 h of incubation [56].

The lowest MIC value of thyme extract was observed in *P. damselae* and *V. vulnificus* at 25 mg/mL. The bacteriostatic concentration of thyme extract on other bacteria was 50 mg/mL. The MBC values of *S. thermophilus* extract were >100 mg/mL, except for *P. damselae* and *V. vulnificus* bacteria. The bactericidal concentration of thyme extract was 50 mg/mL for *Vibrio vulnificus*, *Enterococus faecalis*, and *Salmonella* Paratyphi A, and 100 mg/mL for other bacteria.

Thyme extract derived from various extraction conditions had the greatest efficiency in suppressing the growth of *E. faecalis* (MIC—0.313 mg/mL) and showed the same MIC value for *S. aureus* and *Y. enterocolitica* (1.25 mg/Ml) [18]. The hexanic extract of thyme had the strongest suppressive action against *Salmonella* Typhimurium, *E. faecalis*, and *E. coli*, with an MIC of 0.25 mg/disc, but strains of Methicillin-resistant *S. aureus* and *S. aureus* were less sensitive, recording 0.50 mg/disc [18]. Variations in the antibacterial activity of the extracts may be connected to the extraction methods and various concentrations of their major and minor components, as well as to the additive effect of all the components and the various microorganisms that were examined [57]. Thyme essential oil exhibited greater bacteriostatic and bactericidal properties against Gram-positive pathogens in comparison to Gram-negative ones [58]. In this study, non-encapsulated thyme extract played an effective role on all Gram-negative and -positive bacteria, although their inhibition dose varied depending on the bacterial strains. Thyme essential oil contains high concentrations of p-cymene, thymol, and γ-terpinene, exerting the ability to obstruct the growth of *Bacillus cereus* by breaking down membranes, changing the shape of cells, and lowering the amount of ATP that is present inside cells [59]. In the current study, the antibacterial action of thyme extract might be attributed to the carvacrol and 2-fluoro-2-methyl- propane it contains.

The MBC values of *S. thermophilus* extract were > 100 mg/mL, except for those of *P. damselae* and *V. vulnificus* bacteria. Similar to thyme extract, the microencapsulation of the CFS from *S. thermophilus* showed a strong effect compared to the crude form, and the MIC value was more than 100 mg/mL against all bacteria except *Vibrio vulnificus* (100 mg/mL). Spray drying was expected to significantly impair the antioxidative activity following microencapsulation because the high temperatures during the microencapsulation procedure altered the major phenolic components of thyme extract [25]. This may explain why microencapsulated samples have lower antimicrobial activity than the non-encapsulated form. Taking into consideration the loss of bioactivity due to the thyme microencapsulation via the spray dryer, it was determined that a concentration of 2% and higher should be employed in order to improve the activity of CFS from *S. thermophilus*. The microencapsulated CFS of *S. thermophilus* containing 1% and 2% thyme extract additives generally had a bacteriostatic concentration of 100 mg/mL against bacteria. However, the microencapsulated CFS of *S. thermophilus* containing a 2% thyme extract additive had a low inhibition concentration against *V. vulnificus*, *Proteus mirabilis*, *S. aureus*, and *S.* Paratyphi A (50 mg/mL). The bactericidal value of microencapsulated samples against all examined bacteria was > 100 mg/mL, with the exception of the microencapsulated CFS from *S. thermophilus* with a 2% (*v*/*w*) thyme extract additive. Similar to this study, Gedikoğlu et al. [16] reported that the inhibition effect of the extract against *Staphylococcus aureus* and *Candida albicans* increased by increasing the thyme extract concentration from 10% to 40%. When EOs and bacteriocins are used together, they can create membrane pores that change the permeability of the membrane, the proton motive force, the amino acid efflux, and the pH gradient of the bacterium. Nisin and thyme essential oils together demonstrated synergistic effects against *S.* Typhimurium, although individual thyme essential oil was ineffective in the growth of this bacterium [60]. In this study, combinations of bacterial CFS and 2% (*v*/*w*) thyme extract showed the strongest synergistic impact in preventing bacterial growth in microencapsulated samples.

## 4. Conclusions

The study’s findings showed that microencapsulated CFS from *S. thermophilus* together with thyme extract had a strong antibacterial impact, though their effects on food-related bacteria varied depending on the species of bacteria. In microencapsulation, the combined use of thyme extract with CFS from *S. thermophilus* showed a potent antimicrobial effect on food-related bacteria. Therefore, these microencapsulated extracts have the potential to be employed as alternative antimicrobials in foods. In the microencapsulation of CFS from lactic acid bacteria, the thyme extract additive showed an increasing effect on the antimicrobial properties of CFS. However, considering the results obtained in the study and the losses in bioactive substances in the microencapsulation of thyme extract with a spray dryer, it is recommended that 2% (*v*/*w*) and higher concentrations be studied in further studies to obtain better results regarding their bioactivity. 

## Figures and Tables

**Figure 1 foods-13-00329-f001:**
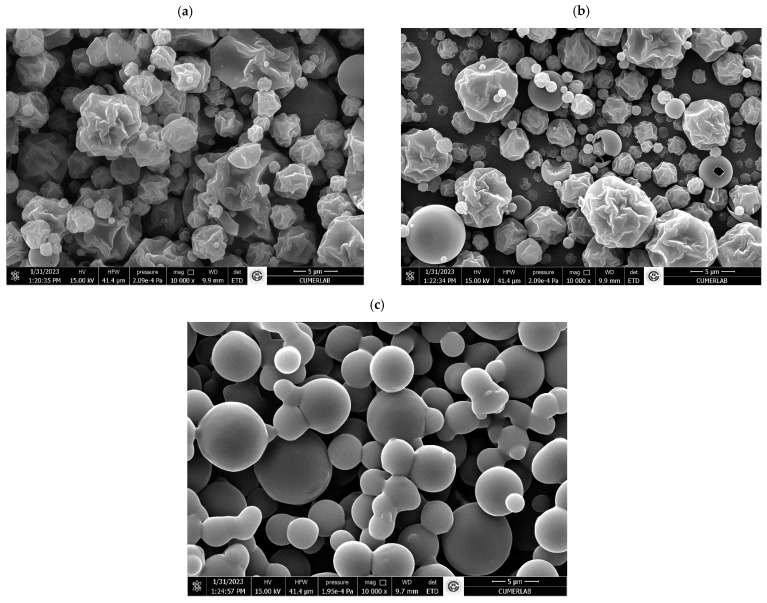
SEM analysis of encapsules. SEM analysis of cell-free supernatant (CFS) of *S. thermophilus* encapsules. (**a**) Microencapsulated CFS from *S. thermophilus* (**b**) microencapsulated CFS from *S. thermophilus* combined with thyme extract (1%, *v*/*w*), (**c**) microencapsulated CFS from *S. thermophilus* combined with thyme extract (2%, *v*/*w*).

**Figure 2 foods-13-00329-f002:**
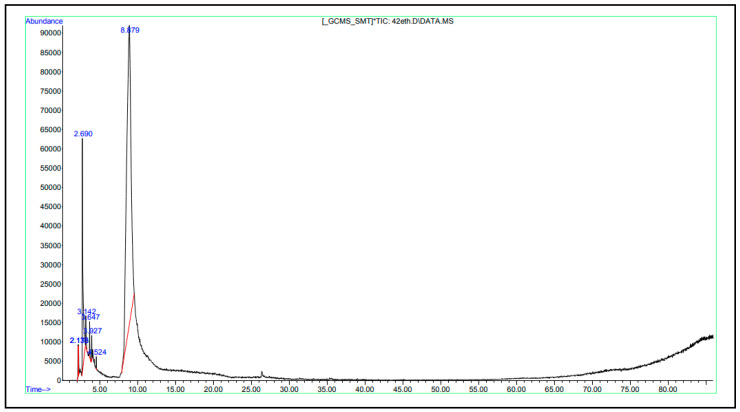
GC-MS spectra of volatile composition of CFS from *S. thermophilus*.

**Table 1 foods-13-00329-t001:** Chemical content of the CFS from *Streptococcus thermophilus*.

Compounds	RT	%
Methyl-d3 1-dideuterio-2-propenyl ether	2.125	10.87
7-Octen-2-ol, 2-methyl-6-methylene	2.176	10.46
Acetic acid	2.686	39.64
Imidodicarbonic diamide	3.143	7.64
Tetrahydro-1,3-oxazine-2-thione	3.647	7.21
2-methyl-1,3-oxathiolany-propionicacid ethyl ester	3.927	5.42
Butanoic acid, 3-methyl- (isovaleric acid)	4.522	0.28
(*R**,*S**)-2-(1′-Nitroethyl)-2-methyl- 1,3-oxathiolane	7.978	0.94
Methanamine, N-methoxy-	8.470	2.82
Propane, 2-fluoro-2-methyl-	8.636	8.92
1,2,3-Propanetriol	8.762	5.8

RT: retention time.

**Table 2 foods-13-00329-t002:** Chemical content of thyme extract.

Compounds	RT	%
Ethane, 1,1-diethoxy-	3.149	0.04
1,1-Diethoxy-2-methylpropan-2-ol	7.749	0.07
1,2,3,4-Tetrahydroxybutane	8.310	0.04
1,2,3-Propanetriol (Glycerin)	8.362	25.77
Propane, 2-fluoro-2-methyl-	8.499	3.97
Thymol	19.411	0.05
Phenol, 2-methyl-5-(1-methylethyl) (carvacrol)	19.726	67.96
Unidentified	-	2.10

RT: retention time.

**Table 3 foods-13-00329-t003:** Inhibition zones (mm) of extracts towards food-related bacteria.

			Microencapsulated Groups		
	*CFS from S. thermophilus* (CFS)	Thyme Extract (TE)	CFS	CFS + 1%TE	CFS + 2%TE
*Photobacterium damselae*	7.33 ± 0.58 *^b^	25.00 ± 1.41 ^a^	4.50 ± 0.71 ^c^	6.00 ± 0.00 ^bc^	6.00 ± 0.00 ^bc^
*Proteus mirabilis*	15.25 ± 0.71 ^b^	32.00 ± 1.41 ^a^	7.50 ± 0.71 ^d^	7.50 ± 0.71 ^d^	10.50 ± 0.71 ^c^
*Vibrio vulnificus*	17.33 ± 0.58 ^b^	41.67 ± 0.58 ^a^	8.25 ± 1.06 ^e^	11.00 ± 1.41 ^d^	13.50 ± 0.71 ^c^
*Enterococus faecalis*	14.00 ± 1.00 ^b^	25.33 ± 1.53 ^a^	7.00 ± 0.00 ^cd^	7.50 ± 0.71 ^cd^	9.50 ± 0.71 ^c^
*Staphylococcus aureus*	13.25 ± 0.87 ^b^	25.50 ± 1.53 ^a^	6.50 ± 0.50 ^d^	7.25 ± 0.35 ^cd^	8.50 ± 0.71 ^c^
*Salmonella* Paratyphi A	13.50 ± 0.50 ^b^	27.33 ± 1.53 ^a^	6.25 ± 0.25 ^d^	7.00 ± 0.00 ^d^	9.38 ± 0.13 ^c^

* Mean value ± Standard deviation (*n* = 3). There is a significant difference (*p* < 0.05) between the groups for the values indicated by different letters (^a–e^) in the same row. CFS + 1%TE: combination of CFS from *S. thermophilus* and thyme extract (1%, *v*/*w*) microencapsules, CFS + 2%TE: combination of *S. thermophilus* and thyme extract (2%, *v*/*w*) microencapsules.

**Table 4 foods-13-00329-t004:** Minimum inhibitory (MIC) and bactericidal concentration (MBC) of extracts samples against food-related bacteria.

					Microencapsulated Groups					
	*CFS from S. thermophilus* (CFS)		Thyme Extract (TE)		CFS		CFS +1%TE		CFS + 2%TE	
	MIC	MBC	MIC	MBC	MIC	MBC	MIC	MBC	MIC	MBC
*Photobacterium damselae*	100	>100	25	100	>100	>100	100	>100	100	>100
*Proteus mirabilis*	25	100	50	100	>100	>100	100	>100	50	>100
*Vibrio vulnificus*	12.5	100	25	50	100	>100	100	>100	50	100
*Enterococus faecalis*	100	>100	50	50	>100	>100	100	>100	50	100
*Staphylococcus aureus*	100	>100	50	100	>100	>100	100	>100	100	>100
*Salmonella* Paratyphi A	100	>100	50	50	>100	>100	100	>100	50	100

Mean value ± Standard deviation (*n* = 3). CFS + 1%TE: combination of CFS from *S. thermophilus* and thyme extract (1%, *v*/*w*) microencapsules, CFS +2%TE: combination of *S. thermophilus* and thyme extract (2%, *v*/*w*) microencapsules.

## Data Availability

Data are contained within the article.

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
