# Peer review of "Bioactivity of Microencapsulated Cell-Free Supernatant of Streptococcus thermophilus in Combination with Thyme Extract on Food-Related Bacteria"

_foods, 2024, doi:10.3390/foods13020329_

Round 1
Reviewer 1 Report
Comments and Suggestions for Authors
Dear Editors and authors,
1- The introduction in the manuscript needs substantial revisions, including adding a paragraph about the contamination of fish and marine organisms with pathogenic bacteria. It is also necessary to add some references in these paragraphs. Please read and add
Özogul, Y., El Abed, N., & Özogul, F. (2022). Antimicrobial effect of laurel essential oil nanoemulsion on food-borne pathogens and fish spoilage bacteria. Food Chemistry, 368, 130831.
Niamah, A. K. (2012). Detected of aero gene in Aeromonas hydrophila isolates from shrimp and peeled shrimp samples in local markets. The Journal of Microbiology, Biotechnology and Food Sciences, 2(2), 634.
2-The purpose of the manuscript is unclear and lengthy; it should be clarified and shortened.
3-Page 3, line 114 , How many viable bacterial colonies were counted?
4-How do you scientifically explain the lack of effect of thyme extract on bacteria despite the extract containing many phenols, tannins, and other inhibitory substances? The following article can be referenced. Sateriale, D., Forgione, G., De Cristofaro, G. A., Pagliuca, C., Colicchio, R., Salvatore, P., ... & Pagliarulo, C. (2023). Antibacterial and Antibiofilm Efficacy of Thyme (Thymus vulgaris L.) Essential Oil against Foodborne Illness Pathogens, Salmonella enterica subsp. enterica Serovar Typhimurium and Bacillus cereus. Antibiotics, 12(3), 485.
5-Agar Well Diffusion Method, This method is ambiguous and unclear. What is the size of the bacterial vaccine used from pathogenic bacteria? What are the types of bacteria used? How many viable cells are there in the vaccine volume?
6-Modern nomenclature must be used when writing the names of lactic acid bacteria throughout the manuscript , See Lactobacillus plantarum correct to Lactiplantibacillus plantarum.
7-Table 1, the compounds that appeared were selected by the authors and this is a big mistake. Look at the third column, how is it a percentage and the total does not reach 100%??? How is that?
8-Table 2, The same observation in the previous paragraph. Why does the percentage not reach 100%?
9- The conclusions contain multiple findings. The conclusions need to be rewritten.
Author Response
Reviewer: 1
Suggestion /corrections 1: The introduction in the manuscript needs substantial revisions, including adding a paragraph about the contamination of fish and marine organisms with pathogenic bacteria. It is also necessary to add some references in these paragraphs. Please read and add
Özogul, Y., El Abed, N., & Özogul, F. (2022). Antimicrobial effect of laurel essential oil nanoemulsion on food-borne pathogens and fish spoilage bacteria. Food Chemistry, 368, 130831.
Niamah, A. K. (2012). Detected of aero gene in Aeromonas hydrophila isolates from shrimp and peeled shrimp samples in local markets. The Journal of Microbiology, Biotechnology and Food Sciences, 2(2), 634.
Response 1. We are thankful for the improvement of manuscript by the Reviewer. The section was improved according to Referee’s suggestions.
Suggestion /corrections 2: -The purpose of the manuscript is unclear and lengthy; it should be clarified and shortened.
Response 2. The aim section of the study has been revised and rewritten
Suggestion /corrections 3: 3-Page 3, line 114 , How many viable bacterial colonies were counted?
Response 3. In the section mentioned by the referee, bacterial cell free extract (CFS) was used and the sample does not contain any cells. However, the cell dose used when obtaining the supernatant was 108 cfu/mL and, this information was detailed in the section on CFS preparation.
Suggestion /corrections 4. How do you scientifically explain the lack of effect of thyme extract on bacteria despite the extract containing many phenols, tannins, and other inhibitory substances? The following article can be referenced.
Sateriale, D., Forgione, G., De Cristofaro, G. A., Pagliuca, C., Colicchio, R., Salvatore, P., ... & Pagliarulo, C. (2023). Antibacterial and Antibiofilm Efficacy of Thyme (Thymus vulgaris L.) Essential Oil against Foodborne Illness Pathogens, Salmonella enterica subsp. enterica Serovar Typhimurium and Bacillus cereus. Antibiotics, 12(3), 485.
Response 4. In this study, non-encapsulated thyme extract played an effective role on all Gram negative and positive bacteria, although their inhibition dose varied depending on the bacterial strains. However, the microencapsulation forms of these samples had lower activity. This is known to be due to the loss of some bioactive substances by spray drying. These sections have been improved in line with the reviewers' suggestions, taking into account the relevant references.
Suggestion /corrections 5-Agar Well Diffusion Method, This method is ambiguous and unclear. What is the size of the bacterial vaccine used from pathogenic bacteria? What are the types of bacteria used? How many viable cells are there in the vaccine volume?
Response 5. The method was given in detail.
Suggestion /corrections 6-Modern nomenclature must be used when writing the names of lactic acid bacteria throughout the manuscript , See Lactobacillus plantarum correct to Lactiplantibacillus plantarum.
Response 6. The names of lactic acid bacteria were corrected. We are thankful to the referee for the scientific improvement of the article
Suggestion /corrections 7-8. Table 1, the compounds that appeared were selected by the authors and this is a big mistake. Look at the third column, how is it a percentage and the total does not reach 100%??? How is that?
Table 2, The same observation in the previous paragraph. Why does the percentage not reach 100%?
Response 7-8. In Table 1, the sum of the compounds is given as 100%, while Table 2 is given as 97.9%. 3.1% of this table (Table 2)is indicated as unidentified compounds because some compounds could not be identified here.
Suggestion /corrections 9- The conclusions contain multiple findings. The conclusions need to be rewritten.
Response 9. The conclusions section was rechecked and improved.
Reviewer 2 Report
Comments and Suggestions for Authors
The manuscript entitled “ Bioactivity of Microencapsulated Cell Free Supernatant of Streptococcus thermophilus in Combination with Thyme Extract on Food Related Bacteria” is interesting. The authors still need to improve the overall quality of the manuscript and scientific understanding to justify some of the results. The comments are mentioned below:
1. The authors should state the necessary of the thyme extract inclusion in the introduction part.
2. Line 94: The authors should supplement the concentration of the thyme extract.
3. Figure 1: The scale in Figure 1 (b) differs from Figure 1 (a) and (c).
4. GC-MS spectra of the samples should be supplemented.
5. Line 269-270: The authors may try to explain why the microencapsulated samples exhibit lower inhibition zone than non-encapsulated samples.
Author Response
Reviever 2
The manuscript entitled “ Bioactivity of Microencapsulated Cell Free Supernatant of Streptococcus thermophilus in Combination with Thyme Extract on Food Related Bacteria” is interesting. The authors still need to improve the overall quality of the manuscript and scientific understanding to justify some of the results. The comments are mentioned below:
Suggestion /corrections 1. The authors should state the necessary of the thyme extract inclusion in the introduction part.
Response 1. We are thankful for the improvement of manuscript by the Reviewer. Information about thyme extract has been added to the Introduction section.
Suggestion /corrections 2. Line 94: The authors should supplement the concentration of the thyme extract.
Response 2. The extract concentration was added the text
Suggestion /corrections 3. Figure 1: The scale in Figure 1 (b) differs from Figure 1 (a) and (c).
Response 3. The correction was made and the results were only given at 10000 magnification.
Suggestion /corrections 4. GC-MS spectra of the samples should be supplemented.
Response 4. Only the GC-MS chromatogram of the free cell extract from S. thermophilus of this study is available, therefore only this chromatogram was included in text (Fig.1).
Suggestion /corrections 5. Line 269-270: The authors may try to explain why the microencapsulated samples exhibit lower inhibition zone than non-encapsulated samples.
Response 5. Spray drying was expected to significantly impair the bioactivity following microencapsulation because the high temperatures during the microencapsulation procedure altered the major phenolic components of thyme extract (YeÅŸilsu and Özyurt, 2019). This may explain why microencapsulated samples have lower antimicrobial activity than the non-encapsulated form. This explanation was also given in the article
Reviewer 3 Report
Comments and Suggestions for Authors
In this paper,the bioactive properties of the combination of microencapsulated cell free supernatant from Streptococcus thermophilus and thyme extract on fish spoilage and pathogenic bacteria were investigated. From my point of view, the work and the results are worthy of affirmation. However, there are still some problems that need to be modified before the inspection.
1. In 3.1, Why do different concentrations of thyme additives produce wrinkles need to be further explained.
2. In 3.2, as showed in table 1, no lactic acid was detected from S. thermophilus, why? Meanwhile, the author also pointed out that lactic acid bacteria can produce a variety of antibacterial substances in metabolism, such as bacteriocin, etc. However, the author did not conduct more detection and analysis, please explain.
3. Table 2 mainly shows the chemical composition of thyme extracts, but the authors do not explain the relationship between these substances and antibacterial activity.
4. Line 168, the P need italics, please check below.
5. In 3,3, also need more discussion and analysis, the causes of different antibacterial effects, and thyme extract has better antibacterial effect, why not further increase its concentration
6. In terms of antibacterial effect, the sample prepared in this study did not have good antibacterial effect. The author needs to clarify the significance of this study.
Comments on the Quality of English LanguageMinor editing of English language required
Author Response
Reviewer 3
In this paper,the bioactive properties of the combination of microencapsulated cell free supernatant from Streptococcus thermophilus and thyme extract on fish spoilage and pathogenic bacteria were investigated. From my point of view, the work and the results are worthy of affirmation. However, there are still some problems that need to be modified before the inspection.
Suggestion /corrections 1. In 3.1, Why do different concentrations of thyme additives produce wrinkles need to be further explained.
Response 1: The collapses occurring in microencapsules may be due to the possibility of moisture transport during the drying period [24,37]. The wrinkled form in thyme extracts may result from this. This explanation is already given in the article
Suggestion /corrections 2. In 3.2, as showed in table 1, no lactic acid was detected from S. thermophilus, why? Meanwhile, the author also pointed out that lactic acid bacteria can produce a variety of antibacterial substances in metabolism, such as bacteriocin, etc. However, the author did not conduct more detection and analysis, please explain.
Response 2: Some LAB strains were able to degrade lactic acid to acetic acid. Each mole of lactic acid was converted into approximately 0.5 mol of acetic acid, 0.5 mol of 1,2-propanediol, and traces of ethanol (Oude Elferink et al., 2001). This may be the reason for the high proportion of acetic acid (39.64%) in the chemical composition of CFS from S. thermophilus. in the present study, a separate analysis for lactic acid in HPLC was not planned. the referee is right about this. However, in this direction, more specific organic acid analysis is planned to be carried out in the next studies.
Oude Elferink, S. J., Krooneman, J., Gottschal, J. C., Spoelstra, S. F., Faber, F., & Driehuis, F. (2001). Anaerobic conversion of lactic acid to acetic acid and 1, 2-propanediol by Lactobacillus buchneri. Applied and Environmental microbiology, 67(1), 125-132.
Suggestion /corrections 3. Table 2 mainly shows the chemical composition of thyme extracts, but the authors do not explain the relationship between these substances and antibacterial activity.
Response 3: The discussion section was improved according to Referee’s suggestions.
Suggestion /corrections 4. Line 168, the P need italics, please check below.
Response 4: The corrections were made.
Suggestion /corrections 5. In 3,3, also need more discussion and analysis, the causes of different antibacterial effects, and thyme extract has better antibacterial effect, why not further increase its concentration
Response 5: The manuscript was improved according to Referee’s suggestions.
Suggestion /corrections 6. In terms of antibacterial effect, the sample prepared in this study did not have good antibacterial effect. The author needs to clarify the significance of this study.
Response 6. The modifications were made in the text. We are thankful the referee for his/her scientific contributions and improvements to the article
Round 2
Reviewer 1 Report
Comments and Suggestions for Authors
Dear Editors,
The authors made all the corrections required of them in the first review. The manuscript became good and can be published in its current form.
Author Response
We are appreciative of the referee's input regarding the article's scientific advancement.
Reviewer 2 Report
Comments and Suggestions for Authors
This paper can be accepted.
Author Response

(The authors gave the same response as above.)

Reviewer 3 Report
Comments and Suggestions for Authors
The author has revised it as requested.
Author Response

(The authors gave the same response as above.)
